# Improvements in Protoplast Isolation Protocol and Regeneration of Different Cabbage (*Brassica oleracea* var. *capitata* L.) Cultivars

**DOI:** 10.3390/plants12173074

**Published:** 2023-08-27

**Authors:** Ester Stajič

**Affiliations:** Department of Agronomy, Biotechnical Faculty, University of Ljubljana, 1000 Ljubljana, Slovenia; ester.stajic@bf.uni-lj.si

**Keywords:** *Brassica oleracea*, protoplasts, isolation, regeneration

## Abstract

Protoplasts are a versatile tool in plant biotechnology since they can be used for basic biological studies as well as for breeding strategies based on genome editing. An efficient protoplast isolation protocol is essential for conducting protoplast-based studies. To optimize the protoplast isolation protocol in cabbage (*Brassica oleracea* var. *capitata* L.), different enzyme solutions were tested for the isolation of leaf mesophyll protoplasts. In our experiments, the combination of 0.5% Cellulase Onozuka RS and 0.1% Macerozyme R-10 showed the best result. The optimized protocol proved suitable for the isolation of protoplasts from five different cabbage cultivars with yields ranging from 2.38 to 4.63 × 10^6^ protoplasts/g fresh weight (fw) and a viability of 93% or more. After three weeks in culture, protoplasts from all of the tested cultivars formed micro-calli, but further callus growth and shoot regeneration depended strongly on the genotype and regeneration protocol used. For shoot formation, 1 mg/L BAP in combination with auxin 0.2 mg/L NAA showed the best results with a regeneration of 23.5%. The results obtained will contribute to the development of different applications of cabbage protoplasts and facilitate the breeding process of this important horticultural crop.

## 1. Introduction

Cabbage (*Brassica oleracea* var. *capitata* L.) belongs to the family Brassicaceae, which includes several economically important vegetable species grown worldwide. In Europe, *B. oleracea* varieties such as cabbage, broccoli, and kale are gaining interest due to their high nutritional value and human health benefits [1,2]. Cabbage is a biennial plant that is self-incompatible and requires mandatory vernalization, which greatly affects the breeding process. Therefore, new approaches in plant breeding are needed to accelerate the production of new varieties with desirable agronomic traits.

Protoplasts, plant cells without a cell wall, are a unique system that can be used in basic research such as physiological and biochemical studies, and for new breeding strategies based on protoplast transformation or fusion [3,4]. Somatic hybridization by protoplast fusion has been used for the transfer of desirable resistance genes to diseases or other stress factors in many plants that cannot be crossed in a traditional way [5,6,7]. In addition, protoplast-based transient transformation methods that rely on direct DNA uptake using polyethylene glycol (PEG) or electroporation are simple, highly efficient, and useful for numerous cell-based assays including the study of gene expression regulation, signaling pathways, protein interactions, etc. [8]. Moreover, with the advent of genome editing techniques, protoplasts are gaining more interest as they offer many advantages over stable transformation methods using *Agrobacterium tumefaciens*, which are time-consuming and laborious [9]. Protoplast transformation has been used as a rapid screening method to validate genome editing reagents in many plant species [10,11,12,13,14]. Moreover, they can also be regenerated into whole plants with targeted modifications [13,14,15].

However, in order to apply protoplast technology in practice, the successful isolation of protoplasts is first required. Methods for the enzymatic isolation of protoplasts using cell wall degrading enzymes such as cellulases and pectinases were introduced by Cocking in 1960 [16]. The most commonly used enzymes are Cellulase Onozuka RS and Pectolyase Y-23; the latter can also be replaced by Macerozyme R10 [17]. For the optimal digestion of plant material, various factors such as the choice of enzymes and their concentrations, the duration of digestion, and the condition of the donor material must be tested [3,8,17,18]. The most common donor material is leaves, but protoplasts can also be isolated from petals or seedling organs such as hypocotyls or cotyledons [18,19]. In *B. oleracea* L., protoplasts have been isolated from leaves and hypocotyls using different enzyme solutions. Commonly, Cellulase Onozuka RS at concentrations of 0.1% to 1% is used in combination with 0.1–1% Macerozyme R10 or 0.1% Pectolyase Y-23. Lower yields (1.8 × 10^4^ to 0.8 × 10^6^ protoplasts/g fw) have been reported for hypocotyls than for leaves (1.3 to 3.2 × 10^6^ protoplasts/g fw) [20,21,22,23]. The latter are a good source of obtaining a high number of uniform cells. Moreover, when using in vitro plants, protoplasts can be isolated from micropropagated clones grown under controlled conditions, thus minimizing the negative effects of poor physiological conditions on isolation efficiency [18].

Isolated protoplasts can be used in different transient assays, or if a regeneration protocol is available, plants with improved traits can be obtained by protoplast fusion or transformation. However, protoplast regeneration is a very complex process involving many steps such as cell wall recovery, cell cycle re-entry, callus formation, and de novo tissue regeneration [24]. It also depends on many factors including the protoplast isolation protocol, culture conditions, concentration, and combination of plant growth regulators in culture media, etc., which need to be optimized to achieve plant regeneration from protoplasts [3,18,19,25]. In the initial phase, high osmotic pressure is required to maintain the protoplast’s integrity before the cell wall is rebuilt, and growth regulators such as auxins and cytokinins are essential for protoplast division. However, during culture, the requirements of protoplasts may change, and the composition of the medium must be adjusted [3,26]. For this reason, a liquid medium is usually used for the culture of protoplasts. Liquid media based on Murashige and Skoog [27] or Kao and Michayluk [28] with modifications are usually used for the cultivation and regeneration of Brassica protoplasts [23,29,30,31,32,33]. To prevent the agglutination of protoplasts and promote mitotic divisions, protoplasts can be immobilized in agarose or alginate [3,18,19], an approach often used for *B. oleracea* L. protoplasts [20,22,23,29].

Several protocols for successful protoplast regeneration have been published, mainly for the Solanaceae and Poaceae families [24,34,35,36]. In Brassicaceae, protoplast research has mainly focused on *B. napus* L. due to its importance in oilseed production [8,26,37,38,39]. Sahab et al. [8] published a detailed protocol for the isolation of leaf mesophyll protoplasts of *B. napus* and their regeneration after PEG-mediated transformation. Although there are also publications on cabbage, plant regeneration remains limited to a few cultivars [23,29]. Recently, the effects of supplements such as peptidyl growth factors (phytosulfokines) and polyamines in the culture medium on the division frequency of protoplasts have been studied, but despite the modifications, plant regeneration from cabbage protoplasts remains low [30,31,32].

In this study, factors affecting protoplast regeneration such as the composition of the enzyme solution (concentration and choice of cell-wall digesting enzymes) for successful protoplast isolation and different culture protocols were tested. In addition, the protoplast-to-plant regeneration potential in five different cabbage cultivars was investigated.

## 2. Results

### 2.1. Optimization of Protoplast Isolation Protocol

To determine the optimal composition of the enzyme solution, different concentrations of Pectolyase Y-23 (0.01, 0.03, 0.05, and 0.1%) with 0.5% Cellulase Onozuka RS were used in the overnight digestion of ‘Rebecca F1’ leaves. The released protoplasts were purified by centrifugation on a sucrose gradient to remove debris, and intact protoplasts were collected in interphase. Isolation efficiency was determined by counting protoplasts under the microscope and by FDA staining. No differences in the yield of isolated protoplasts were observed between the different concentrations of Pectolyase Y-23, but the viability of isolated protoplasts decreased dramatically (from 97% to 37%) when the concentration of Pectolyase Y-23 was increased from 0.05% to 0.1% (Table 1, Figure 1).

Next, the isolation efficiency of two different enzymes, Pectolyase Y-23 and Macerozyme R-10, at concentrations of 0.01% and 0.1%, respectively, was compared on the leaves of all of the tested cultivars (‘Rebecca F1’, ‘Reball F1’, ‘Krautman F1’, ‘Primero F1’, and ‘Huzaro F1’). Substitution of Pectolyase Y-23 with Macerozyme R-10 had a positive effect on the protoplast isolation efficiency in all cultivars. After overnight incubation in 0.1% Macerozyme R-10, the plant material was digested more than with 0.01% Pectolyase Y-23. Differences were also observed after centrifugation with the sucrose gradient (Figure 2). On average, the yield was 2.4 times higher with 0.1% Macerozyme R-10 than with 0.01% Pectolyase Y-23. Among the cultivars, the highest isolation efficiency was obtained in ‘Reball F1’ (4.63 ± 0.47 × 10^6^ protoplasts/g fw) (Table 2, Figure 3). High protoplast viability (90% and above) was observed in all cultivars. No differences in protoplast viability were observed between the enzymes used (Figure 4). According to these results, 0.5% Cellulase Onozuka RS with 0.1% Macerozyme R-10 was selected as the optimal combination for protoplast isolation and used in the protoplast regeneration experiments.

### 2.2. Protoplast Cultivation and Regeneration

To improve the regeneration of cabbage protoplasts, different culture media and protocols were tested for the protoplasts isolated with optimized enzyme solution: a protocol proposed by Kiełkowska and Adamus [23] with two variants (Protocols 1 and 2) and a protocol by Jie et al. [33] (Protocol 3). The first cell divisions could be observed 8 days after isolation (Figure 5C). Multicellular colonies became visible in the third week of cultivation when the protoplasts were cultured according to Protocols 1 and 2, while protoplasts cultured according to Protocol 3 regenerated slowly and colonies became visible to the naked eye in the fourth week of cultivation. Four weeks after isolation, the success of micro-calli formation was assessed (Table 3). Micro-calli was obtained from the protoplasts isolated from all cultivars, but differences in the number and size of the micro-calli tissue formed were observed among the cultivars and regeneration protocols used. High micro-calli induction was observed in the ‘Reball F1’ and ‘Huzaro F1’ cultivars with Protocols 1 and 2. Both cultivars formed a high density of micro-calli (Figure 5D), in contrast to ‘Rebecca F1’ and ‘Krautman F1’, which produced only a few micro-calli colonies per alginate layer. The lowest response was observed in cultivar ‘Primero F1’, which formed micro-calli only with Protocol 3. No differences were observed between Protocols 1 and 2, while the micro-calli obtained with Protocol 3 was smaller and did not proliferate further.

On the shoot regeneration medium, micro-calli of the ‘Reball F1’ and ‘Huzaro F1’ protoplasts continued to grow (Figure 5E), but later turned brown and produced only roots. Although ‘Krautman F1’ produced micro-calli with low density, shoots were produced on a regeneration medium containing 1 mg/L BAP and 0.2 mg/L NAA or 1 mg/L BAP only. Higher regeneration (23.5%) was observed when 0.2 mg/L NAA was added to the regeneration medium in Protocol 1 (Table 4).

## 3. Discussion

Traditional breeding of *B. oleracea* varieties has led to significant improvement in productivity and quality [40,41], but to overcome the limitations of current breeding programs and meet the needs of an ever-growing population, the application of novel technologies is necessary. Many protoplast-based approaches have already been used in plant research including genome editing using CRISPR/Cas9 [4,25]. Nevertheless, there are obstacles associated with protoplast regeneration, and in many plant species, efficient protoplast regeneration remains a technical barrier [4]. In addition, isolating a sufficient amount of highly viable protoplasts is also a challenge in some plant species [42].

In protoplast isolation, the selection of enzymes and their concentration play a crucial role as they directly affect the efficiency and viability of the isolated protoplasts [3,18,43]. To obtain a high yield of viable cabbage protoplasts, the effect of different concentrations of Pectolyase Y-23 on the quantity and quality of isolated protoplasts was first tested. Leaves of sterile in vitro plants were used in all experiments to reduce the negative effects of uncontrolled growth conditions and the presence of endogenous bacteria, as suggested by Moon et al. [43]. While there were no significant differences in the yield of isolated protoplasts, higher concentrations of Pectolyase Y-23 affected the viability. Increasing the enzyme concentration can result in higher yields, but excess enzyme can cause phytotoxicity and consequently reduce the viability [44,45], which was also evident in our experiments.

Macerozyme R-10 instead of Pectolyase Y-23 was then used in enzymatic solution and found that the replacement of Pectolyase Y-23 with Macerozyme R-10 improved the isolation efficiency by more than twofold to 4.63 × 10^6^ protoplasts/g fw. Similar results were reported by Kiełkowska and Adamus [23], but they obtained a lower yield (2.1 × 10^6^ protoplasts/g fw) with a higher (1%) concentration of Cellulase Onozuka RS. The enzyme solution optimized in this study with Cellulase Onozuka RS and Macerozyme R-10 at concentrations of 0.5% and 0.1%, respectively, was suitable for the isolation of protoplasts with high viability from the leaves of all five different cabbage cultivars. This demonstrates the robustness and wide applicability of the isolation protocol optimized in this study.

In some plant species, higher isolation efficiencies (up to 1 × 10^7^ protoplasts/g fw) were obtained by adjustments to the isolation protocol [46,47]. In *Arabidopsis*, the so-called “Tape-*Arabidopsis* Sandwich” isolation method has been developed, which does not require slicing of the leaves, which is time-consuming and can damage the cells. Instead, the epidermis is peeled off with tape and the mesophyll cells are exposed to the digesting enzymes. This approach exposes a larger surface area to the enzymes, resulting in the release of a larger amount of protoplasts and a shorter isolation procedure [46]. This protocol was also applied to other Brassicaceae species such as *B. oleracea* and *B. napus* by Lin et al. [24], but the yield of the obtained protoplasts was not disclosed. Nevertheless, protoplasts were used for PEG transformation, and successful genome editing by CRISPR/Cas9 was reported.

Purification of viable protoplasts by sucrose gradient centrifugation was used in all of the experiments to remove the cell wall debris and dead cells that could have negative effects and inhibit protoplast division and development [25,43]. To prevent the aggregation of protoplasts during culture, they were immobilized in alginate layers right after isolation, which has been reported to promote cell division and improve the plating efficiency [23,43,48,49]. 

The composition of protoplast culture media and the culture protocol are key factors influencing the division of protoplasts and their regeneration into plants. For this reason, three different protocols for the culture of cabbage protoplasts that differ in osmotic and plant growth regulators were examined. The latter are crucial in initial protoplast culture [25,26]. In all of the protocols tested, 2,4-D was used as recommended for Brassicaceae protoplasts [50]. 2,4-D was also essential for the cell wall formation and initial growth of *B. napus* protoplasts [26]. Cell divisions of protoplasts of all cultivars could be observed in the first two weeks, but micro-calli from the ‘Primero F1’ protoplasts was produced only by Protocol 3, while Protocols 1 and 2 were more suitable for micro-calli induction in all other cultivars. The choice of osmotic stabilizer may have had an influence here. In Protocol 3, myo-inositol was used together with sucrose, as opposed to glucose, the main osmotic stabilizer in Protocols 1 and 2. Myo-inositol was proposed by Jie et al. [33] as an osmotic regulator in the culture of cabbage protoplasts, but proved to be unsuitable in our experiments as only a low-density micro-calli was obtained, which did not proliferate further. In Protocol 2, the osmotic and plant growth regulators were gradually changed during cultivation, but no differences in the efficiency of micro-calli induction could be observed compared to Protocol 1. 

It has been shown several times that the genotype plays a significant role in the response of *B. oleracea* protoplasts [23,29,30,31]. In this study, all of the tested cultivars produced micro-calli, but only protoplasts of ‘Krautman F1’ formed shoots. It has been reported that the optimal protoplast culture medium may vary among cultivars [19]. Surprisingly, in our experiments, no shoots were obtained from the protoplasts of ‘Reball F1’, which is considered one of the responsive cabbage cultivars [23,30]. One of the possible reasons could be the too high density of protoplasts. Andersson et al. [35] and Moon et al. [43] reported that too low or too high protoplast density can inhibit shoot regeneration. This hypothesis could be confirmed since ‘Reball F1’ formed micro-calli with high density but no shoots, while shoots were obtained from the cultivar ‘Krautman F1’, which formed micro-calli with lower density. Micro-calli obtained from ‘Reball F1’ and ‘Krautman F1’ also differed in size, which was inversely proportional to their number. In cauliflower protoplasts, micro-calli of at least 2 mm in size transferred to a solid medium for successful shoot regeneration [51]. 

Shoot regeneration usually requires lower auxin and higher cytokinin levels in Brassicaceae protoplasts [26], while Kiełkowska and Adamus [23] used only cytokinin or even a medium without growth regulators [31]. The addition of auxin (0.2 mg/L NAA) rather than using only cytokinin BAP had a better effect on shoot formation in our experiments when protoplasts were cultured according to Protocol 1. To further optimize the shoot regeneration from cabbage protoplasts, different growth regulators could be tested. Li et al. [26] used ten different combinations and obtained plants in four of them with the highest shoot regeneration of 45.0% on medium supplemented with 2.2 mg/L of TDZ and 1.0 mg/L NAA.

## 4. Materials and Methods

### 4.1. Donor Plant Material

Five different cabbage cultivars (*Brassica oleracea* var. *capitata* L.) were used in this study including four red ‘Rebecca F1’ (Syngenta, Basel, Switzerland), ‘Reball F1’ (Syngenta), ‘Primero F1’ (Bejo, Warmenhuizen, The Netherlands), ‘Huzaro F1’ (Bejo), and one white ‘Krautman F1’ (Bejo). Seeds of all cultivars were surface sterilized by soaking in a 1.66% (*w/v*) solution of dichloroisocyanuric acid (Sigma, St. Louis, MI, USA) for 15 min, followed by three washes in sterile distilled water. Murashige and Skoog medium [27] (Duchefa, Haarlem, The Netherlands) supplemented with 3% sucrose (Duchefa) and 0.8% agar (Duchefa) was used for seed germination. After one week, the seedlings were placed in sterile ECO2 plastic vessels (Duchefa) containing the same medium and kept in a growth chamber with a 16-h-light/8-h-night photoperiod at 21 °C for 6 weeks.

### 4.2. Protoplast Isolation

The newly developed leaves (approximately 1 g) of all five cultivars were cut into small pieces with a sharp scalpel and placed in a preplasmolysis solution (0.5 M mannitol). The material was incubated at room temperature for one hour, and then the solution was replaced with 8 mL of enzyme solution containing 0.5% Cellulase Onozuka RS (>16,000 u/g; Yakult Pharmaceuticals, Tokyo, Japan), 0.01–0.1% Pectolyase Y-23 (>1000 u/g; Duchefa) or 0.1% Macerozyme R-10 (>3000 u/g; Duchefa), 2 mM MES (pH 5.7), 3 mM CaCl_2_, and 0.4 M mannitol. The release of protoplasts took place overnight. For the purification of protoplasts, the undigested plant material was removed by filtration through a 40-µm nylon filter, and the protoplasts were collected by centrifugation at 900 RPM for 5 min. The pellet of protoplasts was then resuspended in 8 mL of 0.5 M sucrose with 1 mM MES layered by 2 mL of W5 solution (154 mM NaCl, 125 mM CaCl_2_, 5 mM KCl, 2 mM MES; pH 5.7) and centrifuged at 1100 RPM for 10 min. Protoplasts from the interphase were transferred to a new tube and washed with 10 mL W5. Finally, the protoplasts were resuspended in 1 mL of culture medium (Table 5). All liquid media were filter-sterilized using a 0.2 μm filter (TPP Techno Plastic Products AG, Trasadingen, Switzerland).

To determine the yield of the isolated protoplasts, they were counted with a hemocytometer. The viability of protoplasts was assessed by staining with fluorescein diacetate (FDA) [52]. The final density of protoplasts was adjusted to 8 × 10^5^/mL. Data obtained from different isolation experiments were analyzed using the Student’s *t*-test or ANOVA and Tukey’s test. Detailed information is provided at the end of the corresponding tables and figures. All data are expressed as the mean ± standard error (SE).

### 4.3. Protoplast Regeneration

For the regeneration of protoplasts, they were embedded in thin alginate layers as previously described by Kiełkowska and Adamus [23] with some modifications. An equal volume of alginate solution consisting of 2.8% sodium alginate (Sigma) and 0.4 M mannitol, pH 5.8 was added to the protoplast suspension with an adjusted density of protoplasts to 8 × 10^5^ p/mL and mixed gently. A total of 300 µL of the mixture was spread onto 1% agar (Duchefa) containing 20 mM CaCl_2_ and 0.4 M mannitol and left for 1 h until thin layers were formed. The alginate layers were cultured in 6 mL of CPP (Protocols 1 and 2) or J1 (Protocol 3) liquid medium (Table 5) in 55-mm Petri dishes and kept at 25 °C in the dark. After 10 days, the CPP medium was replaced with fresh CPP medium (Protocol 1) or a 1:1 mixture of CPP and CPPD (Protocol 2). Four days later, they were transferred to light. The culture medium was replaced 20 days after isolation with fresh CPP (Protocol 1) or 6 mL CPPD (Protocol 2). In the case of J1 medium (Protocol 3), protoplasts were kept in the dark throughout the period of micro-calli induction. Fourteen days after isolation, the J1 medium was replaced with the J2 medium. Four weeks after protoplast isolation, micro-calli development from the protoplasts was determined using a three-step descriptive scale: no micro-calli tissue (−), low density of micro-calli tissue (+), or high density of micro-calli tissue (++). For plant regeneration, alginate layers with micro-calli were transferred to the MS solid medium supplemented with 3% sucrose, 0.5 or 1 mg/L 6-benzylaminopurine (BAP), and 0.2 mg/L α-naphthaleneacetic acid (NAA) or 2 mg/L 2,4-dichlorophenoxyacetic acid (2,4-D) (Table 6). Plates were maintained in a growth chamber at 25 °C with a 16-h light/8-h night photoperiod.
plants-12-03074-t005_Table 5Table 5Composition of the cultivation media for the cabbage protoplasts used in this study.Medium Name Medium Composition Reference ProtocolCPP Kao and Michayluk [28] macro- and microelements and organic acids, B5 (Gamborg et al. [53]) vitamins, 74 g/L glucose, 250 mg/L casein enzymatic hydrolysate, 0.1 mg/L 2,4-D, 0.2 mg/L zeatin; pH 5.6 [23]1, 2CPPD Kao and Michayluk [28] macro- and microelements and 0.4X organic acids, B5 (Gamborg et al. [53]) vitamins, 20 g/L mannitol, 30 g/L sucrose, 250 mg/L casein enzymatic hydrolysate, 0.1 mg/L NAA, 0.2 mg/L zeatin; pH 5.6 [49]2J1 MS medium NH_4_NO_3_ free (#M0238, Duchefa), 30 g/L sucrose, 60 g/L myo-inositol, 0.4 mg/L thiamine-HCl, 2 mg/L 2,4-D, 0.5 mg/L BAP; pH 5.8 [33]3J2 MS basal macro- and micronutrients (#M0221, Duchefa), 30 g/L sucrose, 60 g/L myo-inositol, 0.4 mg/L thiamine-HCl, 2 mg/L 2,4-D, 0.5 mg/L BAP; pH 5.8 [33]3

## 5. Conclusions

In the present study, high-yielding, viable protoplasts from five cabbage cultivars were obtained by optimizing the composition of the enzyme solution. Micro-calli was formed from protoplasts of all five cultivars, but the density of the micro-calli obtained was not only cultivar but also protocol dependent. Plant regeneration was achieved in the cultivar ‘Krautman F1’, while the micro-calli of the other cultivars failed to produce shoots. Further research will therefore focus on optimizing the cultivation protocol to increase shoot regeneration. The obtained results pave the way for the use of cabbage protoplasts and further improvement in this important horticultural plant.

## Figures and Tables

**Figure 1 plants-12-03074-f001:**
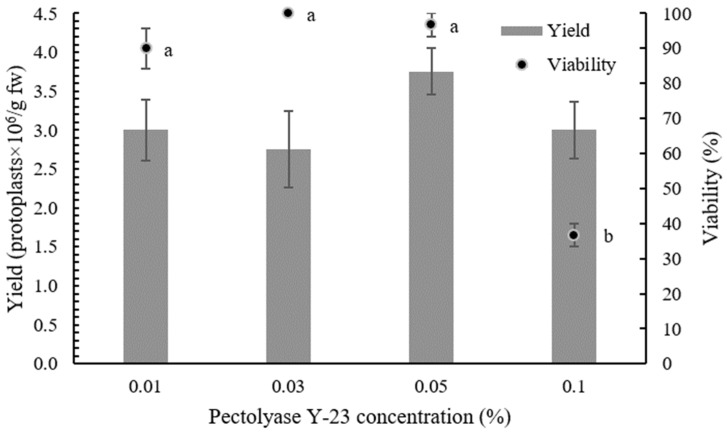
Effect of different concentrations of Pectolyase Y-23 on the isolation of the ‘Rebecca F1’ protoplast. Gray bars present the yield and black dots represent the viability of isolated protoplasts. Values represent the means ± SE. Different letters indicate significant differences at *p* < 0.05.

**Figure 2 plants-12-03074-f002:**
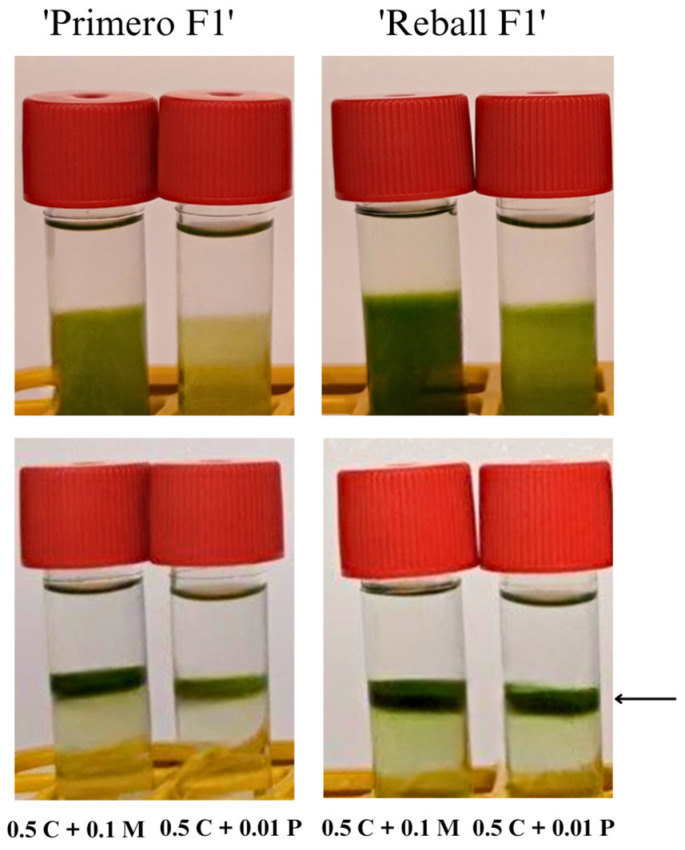
Sucrose gradient purification of protoplast isolated from ‘Primero F1’ and ‘Reball F1’ leaves with 0.01% Pectolyase Y-23 (P) or 0.1% Macerozyme R-10 (M) with 0.5% Cellulase Onozuka RS (C). Protoplasts in 0.5 M sucrose solution overlaid with W5 before (**top**) and after (**bottom**) centrifugation. Arrow indicates the layer of viable protoplasts between both solutions.

**Figure 3 plants-12-03074-f003:**
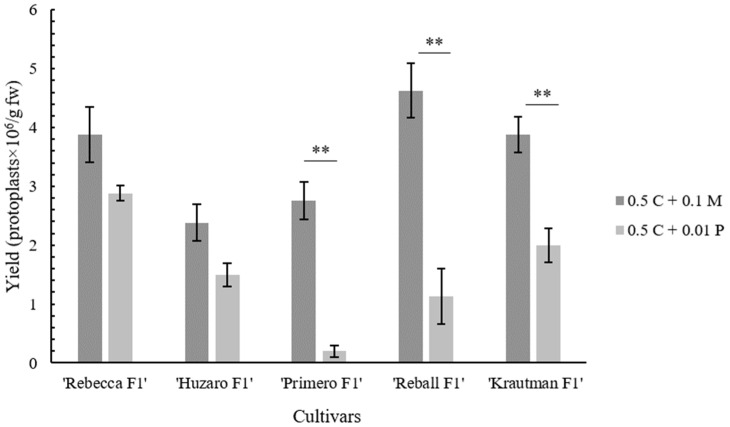
The protoplast isolation efficiency of all of the tested cultivars with 0.01% Pectolyase Y-23 (P) or 0.1% Macerozyme R-10 (M) with 0.5% Cellulase Onozuka RS (C). *p*-values were obtained using the Student’s *t*-test. **, *p* < 0.001.

**Figure 4 plants-12-03074-f004:**
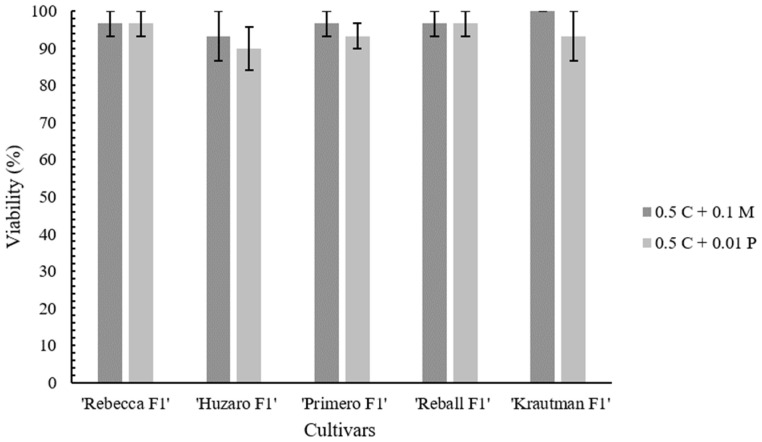
Viability of the protoplasts isolated from the leaves of all of the tested cultivars using 0.01% Pectolyase Y-23 (P) or 0.1% Macerozyme R-10 (M) with 0.5% Cellulase Onozuka RS (C).

**Figure 5 plants-12-03074-f005:**
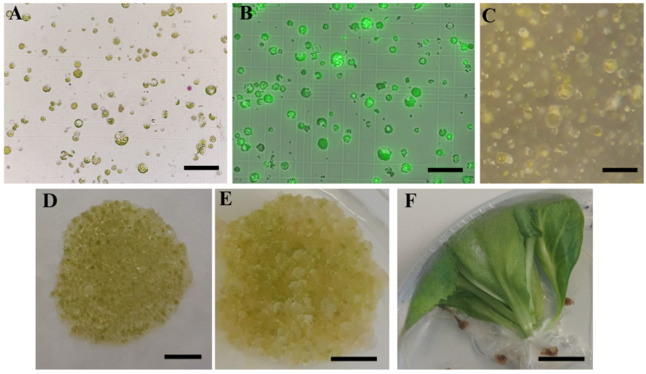
Plant regeneration from the leaf-derived cabbage protoplasts. (**A**) Freshly isolated protoplasts; (**B**) viable protoplast stained with FDA after sucrose purification; (**C**) first cell divisions after 8 days after isolation; (**D**) micro-calli formation after 4 weeks of culture; (**E**) further callus growth after 6 weeks of culture; (**F**) shoot regeneration. Scale bars: 100 µm (**A**–**C**), 1 cm (**D**–**F**).

**Table 1 plants-12-03074-t001:** Comparison of the protoplast isolation efficiency with different concentrations of Pectolyase Y-23.

Cellulase Onozuka RS Concentration (%)	Pectolyase Y-23 Concentration (%)	Yield ± SE (Protoplasts × 10^6^/g fw) ^a^
0.5	0.01	3.00 ± 0.39 ^a^
0.5	0.03	2.75 ± 0.49 ^a^
0.5	0.05	3.75 ± 0.30 ^a^
0.5	0.1	3.00 ± 0.37 ^a^

^a^ Mean values followed by the same letter were not significantly different at *p* = 0.05.

**Table 2 plants-12-03074-t002:** Comparison of the protoplast isolation efficiency (protoplasts × 10^6^/g fw) in different cultivars with 0.01% Pectolyase Y-23 (P) or 0.1% Macerozyme R-10 (M) with 0.5% Cellulase Onozuka RS (C).

Cultivar	Enzyme Solution Composition	
	0.5 C + 0.1 M	0.5 C + 0.01 P
‘Rebecca F1’	3.88 ± 0.47	2.88 ± 0.13
‘Primero F1’	2.75 ± 0.32	0.20 ± 0.10
‘Reball F1’	4.63 ± 0.47	1.13 ± 0.47
‘Krautman F1’	3.88 ± 0.31	2.00 ± 0.29
‘Huzaro F1’	2.38 ± 0.31	1.50 ± 0.20

**Table 3 plants-12-03074-t003:** Micro-calli formation—micro-calli tissue has not formed (−), low density of micro-calli tissue (+) or high density of micro-calli tissue (++).

Cultivar	Protocol 1	Protocol 2	Protocol 3
‘Rebecca F1’	+	+	−
‘Reball F1’	++	++	+
‘Krautman F1’	+	+	−
‘Primero F1’	−	−	+
‘Huzaro F1’	++	++	+

**Table 4 plants-12-03074-t004:** Shoot regeneration from the ‘Krautman F1’ protoplasts (*n* = 17).

Protocol	Medium	PGR Concentration (mg/L)		Regeneration (%)
1	RM11	BAP 1		5.9
1	RM2	BAP 1	NAA 0.2	23.5
2	RM1	BAP 1		0
2	RM2	BAP 1	NAA 0.2	0
3	RM3	BAP 0.5	2,4-D 2	0

**Table 6 plants-12-03074-t006:** Composition of the regeneration media (RM) for the cabbage protoplasts used in this study.

Medium	PGR Concentration (mg/L)		Protocol
RM1	BAP 1		1, 2
RM2	BAP 1	NAA 0.2	1, 2
RM3	BAP 0.5	2,4-D 2	3

## Data Availability

All data are included in the present study.

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
