# Peer review of "Improvements in Protoplast Isolation Protocol and Regeneration of Different Cabbage (Brassica oleracea var. capitata L.) Cultivars"

_plants, 2023, doi:10.3390/plants12173074_

Round 1
Reviewer 1 Report
Dear authors, please find my comments in the attached file.
Best regards,
a reviewer

please have the language checked by a native speaker.
Author Response
Dear Reviewer,
thank you very much for your careful reading and proposed improvements. The suggested changes have been made. Please find point-by-point response in the attachment.
Yours sincerely,
Ester Stajič

Reviewer 2 Report
Review report
General comments: -
This study is very interesting and has a scientific topic with a great impact on the field. The manuscript will be suitable for publication after taking care of the following minor comments.
Detailed comments:
1-The English language and /writing style is fine needs some minor check spelling and grammar check.
2-Please avoid using the personal pronouns (I, We,) such as in line 13 (we in our experiments), in line88 (we focused )and more .
Abstract
_This section is well written
Keywords:
-The keywords has been chosen very carefully and accurately.
Introduction
-The introduction doesn’t provide sufficient background and it is missing enough relevant references
-This section needs to be elongated and enriched with more background about this topic.
Materials and Methods
It is ok and adequate
Results:
The results are very interesting and well presented but figure 5f shoot generation not clear to me.
Discussion:
This section is poorly written
The author is strongly advised to combine the results and discussion in one section for better interpretation and discussion for the presented data especially data in Figure 5 and Table4
Please rewrite and discuss in details, and fully discussed with related citations.
Conclusion
This section is well written and the conclusion is supported by the results of this study and includes the most important findings.
References
This section is well written and UpToDate.
The English and style style is ok just some minor spelling and grammar check up is required.
Author Response

(The authors gave the same response as above.)
